# Brain Network Underlying Executive Functions in Gambling and Alcohol Use Disorders: An Activation Likelihood Estimation Meta-Analysis of fMRI Studies

**DOI:** 10.3390/brainsci10060353

**Published:** 2020-06-07

**Authors:** Alessandro Quaglieri, Emanuela Mari, Maddalena Boccia, Laura Piccardi, Cecilia Guariglia, Anna Maria Giannini

**Affiliations:** 1Department of Psychology, “Sapienza” University of Rome, 00185 Rome, Italy; e.mari@uniroma1.it (E.M.); maddalena.boccia@uniroma1.it (M.B.); laura.piccardi@uniroma1.it (L.P.); cecilia.guariglia@uniroma1.it (C.G.); annamaria.giannini@uniroma1.it (A.M.G.); 2Cognitive and Motor Rehabilitation and Neuroimaging Unit, IRCCS Fondazione Santa Lucia, 00179 Rome, Italy

**Keywords:** pathological gambling, alcohol abuse, decision-making, ALE meta-analysis, functional magnetic resonance imaging, MRI

## Abstract

Background: Neuroimaging and neuropsychological studies have suggested that common features characterize both Gambling Disorder (GD) and Alcohol Use Disorder (AUD), but these conditions have rarely been compared. Methods: We provide evidence for the similarities and differences between GD and AUD in neural correlates of executive functions by performing an activation likelihood estimation meta-analysis of 34 functional magnetic resonance imaging studies involving executive function processes in individuals diagnosed with GD and AUD and healthy controls (HC). Results: GD showed greater bilateral clusters of activation compared with HC, mainly located in the head and body of the caudate, right middle frontal gyrus, right putamen, and hypothalamus. Differently, AUD showed enhanced activation compared with HC in the right lentiform nucleus, right middle frontal gyrus, and the precuneus; it also showed clusters of deactivation in the bilateral middle frontal gyrus, left middle cingulate cortex, and inferior portion of the left putamen. Conclusions: Going beyond the limitations of a single study approach, these findings provide evidence, for the first time, that both disorders are associated with specific neural alterations in the neural network for executive functions.

## 1. Introduction

In the Diagnostic and Statistical Manual of Mental Disorders (DSM-IV) [1] and its revised edition (DSM-IV-TR) [2], pathological gambling (PG) was defined as an impulse control disorder (not otherwise specified) and characterized by “persistent, recurrent and maladaptive behavior that compromises personal, family or work activities” [2]. In contrast with the previous classification, in the DSM-5 [3] Gambling Disorder (GD) is currently included within the spectrum of dependencies and disorders related to substance use (substance-related and addictive disorders). It is described as a chronic and relapsing disorder that affects an individual’s health and sociality, mainly due to addictive behavior (i.e., individuals aim to continue playing despite losing money), loss of control (i.e., despite willingness to stop, they cannot avoid playing again), withdrawal syndrome (i.e., physical and psychological feelings of discomfort when gambling is stopped), and craving (i.e., the compulsive desire to play again).

Several studies have demonstrated the co-occurrence of one or more addictive behaviors (i.e., poly-dependence conditions, namely, different substances and/or gambling addiction in the same individual), as well as cross-dependence (i.e., shifting from one type of addiction to another in the lifespan) [4,5,6]. Accordingly, the current literature on GD has evidenced comorbidity with substance-related abuse and addictive disorders (i.e., alcohol, tobacco, and illegal psychoactive substances), in addition to comorbidity with other diseases, such as depression, hypomania, bipolar disorder; and personality disorders (i.e., antisocial, narcissistic, histrionic, borderline), attention-deficit/hyperactivity disorder (ADHD), panic disorder (with or without agoraphobia); and other medical disorders associated with stress, including peptic ulcer and arterial hypertension [7]. Moreover, several studies investigating the neurophysiological correlates of addictive behaviors and the similarities between drug addiction and behavioral addictions, such as GD, have shown that compulsive behaviors may produce a physiological activation (i.e., arousal) similar to psychoactive substances [8].

Alcohol Use Disorder (AUD) involves deficits in controlling alcohol consumption, extreme concern about alcohol, substance consumption even when it causes problems, the need to drink increasing amounts of alcohol to achieve the same effect, or the presence of withdrawal symptoms (the individual has symptoms such as irritability and emotional outbursts, anxiety, low energy, trouble sleeping, memory problems, dizziness, increased accident proneness, delayed reflexes) when the individual quickly reduces or stops drinking [9]. Harmful alcohol consumption includes the use of alcohol that affects an individual’s health and/or safety, also yielding to other alcohol-related disorders (i.e., alcoholic liver disease, peptic ulcers, sexual dysfunction, cardiovascular disease, and alcoholic dementia). It also includes binge drinking, in which five or more drinks are consumed within two hours. Alcohol abuse can have different effects on different individuals, and many theories suggest that individual factors, such as genetic, psychological, social, and environmental variables, are involved [10]. Alcohol consumption usually begins in adolescence, but alcohol consumption disorder occurs more frequently in the 20s and 30s, although it may begin at any age [9]. The risk factors include drinking steadily over time, early age of onset, family history, depression or other mental health problems, and a history of trauma, as well as social and cultural factors [9].

Following the recent inclusion of GD among the spectrum of dependencies in the DSM-5 [3], a number of studies have attempted to disentangle the common traits of GD and drug addiction. Interestingly, GD and AUD have similar characteristics, such as decision-making deficits and impulsivity, which are also related to relapse and poor therapeutic outcomes. Recent studies on the etiology of GD and AUD report that these two conditions are frequently associated in the same individual (i.e., comorbidity) [11,12,13]. Additionally, GD incidence is up to 10 times higher in alcohol-dependent patients than in the general population [14]. One of the possible reasons for the high comorbidity of AUD and GD is their biological and phenomenological similarity [15]. Specifically, AUD and GD seem to be associated with common clinical features, such as compulsive engagement in a behavior, decreased control over problematic behaviors, craving states, withdrawal symptoms, unsuccessful efforts to stop, and disturbances in the main areas of functioning [16]. Both conditions have been described as sharing a common neurobiological mechanism, which consists of neurofunctional alterations in the ventral tegmental area (VTA), nucleus accumbens (NAc), and orbitofrontal cortex (OFC) [17]. Moreover, both disorders are associated with specific cognitive-emotional mechanisms, including the processing of rewards/punishments and its relation to behavioral conditioning; the increase in the relevance of particular stimuli [18], which often translates into an uncontrolled desire to gamble/drink (craving) [19,20]; and impulsiveness [21,22,23,24], which is considered both a factor of vulnerability to developing addictions and/or the factor that often leads to relapses. Regarding the social context, the prevalence of GD is closely related to the availability and expansion of gambling establishments [25]; this is also true for AUD, since alcohol is usually easily available on the market. Moreover, in contrast to other dependence conditions, such as the use of psychoactive drugs that are stigmatized from the very first encounter, both gambling and alcohol drinking are widely accepted by society as leisure-time activities.

The present literature review suggests that the definition of “executive functions” includes a large umbrella of multiple processes and several, different definition of executive function exist, which refer to different cognitive and neuropsychological models. Therefore, we found that in studying executive functions in GD and in AUD, authors referred to different models and, accordingly, used different tasks to analyze executive functions. The problems of the absence of a homogenous definition of executive function and the large variety of tools used to assess them in the clinical population has already been underlined in other meta-analyses [26].

Here, we provide a revision of the current neuropsychological and neuroimaging literature on AUD and GD, evidencing possible differences or common features of the two conditions. The above-reported literature suggests that both conditions yield alterations in executive functions and the associated brain networks. Thus, we perform a meta-analysis on the functional magnetic resonance imaging (fMRI) studies assessing executive functions (e.g., decision-making, delay discounting, inhibitory response) in individuals with AUD and GD with two main aims, namely (1) summarizing previous findings about the neural correlates of the two conditions in terms of the activation and/or deactivation of the brain areas devoted to executive control, and (2) providing new evidence for common and distinct neural mechanisms in these two conditions.

## 2. Neuropsychological Features of GD and AUD

The pathophysiology of GD, which seems to have common traits with drug addiction [27] and obsessive-compulsive spectrum disorders [28], suggests the possibility of a common psychobiological substrate. GD should result from a strong and complicated interaction between individual vulnerability and environmental factors. This type of behavioral dependence can affect brain functions without the harmful biochemical effects caused by neurotoxic substances. A neuropsychological study of patients with GD identified common traits in neurological patients affected by frontal lobe damage [29], involving an impairment of the decision-making processes; this decision-making deficit leads gamblers to not consider, and thus to ignore, the negative consequences of the immediate reward, inducing them to overestimate the real chances of winning by relying on erroneous beliefs. Patients with GD are more likely to have a reduced sensitivity of the reward system and to be more prone to seek gratifying events. A prevalent symptom in GD is certainly the strong impulse to gamble, which often leads to a relapse [30,31]. Impulsiveness is considered both a factor of vulnerability to developing addictions and a consequence of GD and a deficit in decision-making processes [32]. Potenza and colleagues [33] used the Stroop test to assess neurofunctional alterations associated with deficits in the inhibition of automatic responses in individuals with GD. People with GD showed lower prefrontal activity during the Stroop test compared with healthy controls (HC). GD was associated with both an altered inhibition of impulsive actions and an inadequate ability to delay gratification (impulsive choice) [34], as supported by a number of studies using delay discounting tasks. Delay discounting is an indicator of an individual’s preference for immediate and minor rewards, rather than greater but delayed rewards; it also allows an assessment of the inability to delay rewards. Indeed, an interesting study by Clark and colleagues [35] investigated the possible neuropsychological mechanisms underlying GD, analyzing the “near-miss” trials, in which failures occurred close to the jackpot, thus resulting in an increased desire to play. In a streamlined slot machine scenario, the near winnings were experienced as less pleasant and increased the desire to play compared to full losses. This effect emerged more markedly when the individual actively controlled the bet, supporting the idea that near wins would strengthen the aspects of GD correlated to the desire to play by involving the reward circuit in an anomalous way.

Likewise, patients with AUD have shown a compromised inhibitory process. The effects of alcohol on the brain are different: with low doses, it activates the pleasure areas linked to the release of endorphins; with increasing doses, however, alcohol has a depressive effect on the central nervous system [36]. In particular, it inhibits the function of one of the excitatory neurotransmitters, namely, glutamate, thus slowing down brain activity [36]. The main effects of this inhibition are learning deficits, altered judgment, and lowered levels of self-control [36,37]. Consequently, those who are intoxicated by alcohol often fail to properly assess their abilities. Overall, AUD leads to a progressive loss of judgment and a progressive deterioration of the personality, related to the organic brain damage produced by alcohol [38,39]. This deterioration of personality is manifested by an accentuation of traits, lack of attention and will, loss of interest, emotional disturbances, mood liability, wayward social behavior, reductions in capacity and judgment, and memory loss [39]. The impairment of all these functions can lead to alcoholic dementia [38,39]. Brain damage also occurs in the medium/long term and yields deficits in attentional processes, executive functions, problem solving, decision-making, working memory, and long-term memory [40]. Alcoholic brain damage seems to be partially reversible in some cases, although organic brain damage often remains permanent [40,41]. The neuropsychological consequences of AUD inevitably include a reduced control of behavior and the suppression of psychosocial adaptability skills [42]. Patients with AUD were found to show higher levels of impulsive choice than controls in a delay discounting task; also, higher delay discounting rates were markedly correlated with greater impulsivity trait scores [43]. With regard to delayed decisions, impulsive choices were specifically correlated with higher activity in systems involved in reward processing and monetary evaluation [44]. Abstinent alcohol-dependent individuals showed a greater activity than HC during “sooner-but-smaller” rewards [45]. Lim and colleagues [46] examined whether the severity of AUD was associated with discounting rates on delay discounting tasks, and found that individuals with more severe AUD showed higher discounting rates—namely, a higher tendency to prefer “sooner-but-smaller” to “later-but-larger” rewards. It has also been observed that the performance of individuals with AUD in tasks that use alcohol contextual cues is worse than that of HC [21]. In go/no-go tasks, patients with AUD showed worse performances than controls (both in accuracy and reaction times) during the presentation of alcohol-related images [21]. The alcoholic context may therefore induce a further worsening of the response inhibition deficit in patients with AUD, putting these subjects at risk of making habitual, unwanted responses and relapsing. Decision-making assessed by means of the Iowa Gambling Task (IGT) was worse in patients with AUD, especially during “adapt—choice” behavior; in contrast with HC, patients made a shift in their decisions to disadvantageous cards during the last trials, especially in the last block [47].

## 3. Neurobiological and Neurofunctional Alterations in GD and AUD

Different fMRI studies have analyzed neurofunctional alterations in decision-making processes in individuals with GD. In three studies, the Iowa Gambling Task was used [22,48,49], while three other studies evaluated the decision-making process within a semi-realistic blackjack scenario [50], a card-deck paradigm [51], and a probabilistic gambling scenario [52]. Decreased activity in the ventro medial prefrontal cortex (vmPFC) and in striatal regions of GD individuals has been reported, likely reflecting an orientation to riskier decisions in response to monetary rewards, especially during deck selection [31,33,49,53]. These functional alterations may reflect the poorer performances on the IGT reported above in the “Neuropsychological features of GD and AUD” section, and could lead to decreased contextual learning or decreased sensitivity to monetary punishment correlated with hippocampal and amygdala deactivation [48]. On the other hand, Tanabe and colleagues [22] found a greater reduction in the vmPFC activity in patients with Substance Use Disorder (SUD) and GD comorbidities. This result suggested a relative preservation of vmPFC function in individuals with GD, in contradiction to prior fMRI studies. Scores on the South Oaks Gambling Screen (SOGS) were found to be correlated with the ventral striatum (VS, including nucleus accumbens and olfactory tubercle) activity in individuals with GD [51]. Using an adapted version of the card-deck paradigm, Brevers and colleagues [51] found that patients with GD showed decreased activation in the right globus pallidus during risky decision-making relative to ambiguous decision-making; nonetheless, greater neural activity within the putamen before risky choices than safe choices was reported.

Furthermore, in contrast to HC, patients with GD showed greater activation in the right putamen before high risk choices than before safe choices. Gelskov and colleagues [52] found that, despite similar behavioral performances in individuals with GD and HC (i.e., a similar number of errors in both groups), patients with GD showed greater neural sensitivity in the dorsal cortico-striatal network to the most appetitive and aversive gain–loss ratios. Miedl and colleagues [54], using a delay discounting task, found that those with GD (who showed higher discounting rates) showed an altered functionality in the reward network involving the OFC/VS, the anterior cingulate cortex (ACC) and the VTA. Moreover, the degree of the correlations between the subjective representation of the value and the activity in these areas during the “delay discounting” was directly proportional to the severity of the gambling [55]. Concerning AUD, excessive exposure to the effects of alcohol affects the transmission of signals underlying adaptive learning behavior, assessed in different learning tasks, such as goal-directed vs. habitual learning and instrumental learning [56,57], with an increase in prediction errors, which has been associated with weaker activation of the vmPFC and with an altered connectivity between vs. and prefrontal cortex (PFC) [56,57,58]. This functional alteration has been hypothesized to be associated with a shift from goal-directed actions to habitual actions that, in turn, could reinforce compulsive alcohol consumption and recurrent relapses [59,60,61]. Decreased activity in the vs. has frequently been reported in alcohol-dependent individuals and has often been correlated with the severity of craving, impulsiveness and depressive symptoms [19,62]. However, some studies have reported controversial results, showing an increase in ventral and dorsal striatum activity during tasks of winning or losing anticipation [63], which suggested an impairment in the computation of expected values. A study used a card-guessing task [64] to observe the differences in coding on the expected value of winnings and losses, distinguishing the effects of probability and magnitude of the results: in individuals with AUD, a more pronounced neural response in the left caudate nucleus (CN) was present during the anticipation of winning and in the left CN and left putamen in relation to the magnitude of the results. A general imbalance in the conflict-monitoring brain circuitry underlying decision-making under risk has been described in AUD: specifically, patients with AUD showed enhanced anticipatory activity in neural circuitries of executive and motor control, encompassing the insula (INS), the ACC and the basal ganglia [58,65,66]. In addition, recent studies have demonstrated that the dorso lateral prefrontal cortex (DLPFC) may be particularly hyperactive in individuals with AUD during delayed and cognitively difficult decisions [67]. This activity is associated with cognitive control, suggesting that patients with AUD recruit these brain regions to a greater extent when they make the delayed choice [46,68]. Additionally, brain areas in the parietal and occipital lobes, including the precuneus, angular gyrus and occipital cortex, were more activated during impulsive decisions than delayed choices [46,68]. These regions have been widely reported in the delay discounting literature [44] and are involved in spatially directed attention, multisensory perception and visual processing [69]. McClure and colleagues [70,71] suggested the presence of two separate systems for immediate evaluations and delayed rewards: parts of the limbic system associated with the midbrain dopamine system, including the paralimbic cortex, are preferably driven by decisions involving immediately available rewards, and in contrast, the lateral prefrontal cortex and posterior parietal cortex are engaged by intertemporal choices irrespective of delay.

## 4. Methods

### 4.1. Inclusion Criteria for Papers

We used a systematic approach to review the literature and select appropriate articles for the present meta-analysis. The search for relevant literature was performed using PubMed and Web of Knowledge (Web of Science); the data included in this meta-analysis have been collected from early 2019 to February 2020 and the search was run simultaneously on these databases, using the following keywords and their combinations: fMRI AND “Gambling” OR “Alcohol” AND “Decision Making”, AND “Gambling disorder”, OR “Alcohol Abuse Disorder”, OR “MRI”, OR “Functional MRI”. This search produced 2461 papers. There were 2152 total records remaining after the duplicates were removed.

These articles were then screened according to our a priori inclusion criteria for papers, which were as follows: (1) whole-brain activity analysis performed using functional magnetic resonance imaging (fMRI); thus, we excluded positron emission tomography (PET) studies and excluded papers reporting only results from region of interest (ROI) analyses; (2) the coordinates of the activation foci were provided either in the Montreal Neurological Institute (MNI) or Talairach [72] reference space; the participants included (3) adults who fulfilled the criteria for AUD or GD—hereafter called Groups—without comorbidity or dependence on other substances and (4) HC as a reference for the contrast studies; and (5) the experimental task required participants to perform a decision-making task or similar tasks that engage decisional processes. Studies were included if they met the above inclusion criteria. We followed the guidelines outlined in the Preferred Reporting Items for Systematic Reviews and Meta-analyses (PRISMA) Statement [73,74]. Independent double-checking was used to avoid the biases in the data extraction. Differently from the effect-size meta-analysis, the neuroimaging meta-analysis allowed us to test for the spatial convergence of effects across experiments [75]. Moreover, the strict inclusion criteria of the study defined at the beginning allowed us to check and control for some of these biases. In addition, the same independent investigators have provided the correctness of the space (MNI or Talairach) and the coordinates. Using these criteria, we selected 34 peer-reviewed journal articles (these studies are summarized in Table 1 and Table 2).

The flowchart of article selection is shown in Figure 1. Further details about the study classification are reported below in the “Activation Likelihood Estimation” section.

### 4.2. Activation Likelihood Estimation

The meta-analysis was performed using the activation likelihood estimation (ALE) approach [96,97,98]; ALE is one of the most commonly used techniques for the coordinate-based meta-analysis of neuroimaging data and is found in the GingerALE 3.0.2 software (http://brainmap.org/ale). It is a powerful tool to integrate the neuroimaging literature where relevant regions of activations are identified across a series of studies. Specifically, peak coordinates are obtained from studies that share a common feature of interest, such as specific tasks or processes. The ALE algorithm has been applied in many aspects of normal brain function [99,100,101], as well as in neuropsychiatric and neurological disorders [102,103]. A central feature of the ALE algorithm is to model the coordinates given as the centroids of Gaussian 3-D probability distribution [104], thus embracing the spatial uncertainty of neuroimaging results caused by interindividual neuroanatomical variability and the limits of the signal-noise and spatial resolution of neuroimaging methods. Other algorithms for the meta-analysis of fMRI data do exist—for example, the traditional meta-analysis on mean %BOLD signal change within region of interest (ROI) or Intensity-Based Meta-Analysis (IBMA) on the activation map. However, the former may be affected by different strategies used to identify the ROI in different studies (e.g., atlas-based ROI, using an independent localizer scan, and individual or group ROI); instead, the latter inevitably restricts the number of studies to be included in the meta-analysis, because activation maps are provided very rarely. Thus, a coordinate-based meta-analysis, such as the ALE meta-analysis, offers a good tool to summarize previous findings and generate new hypotheses [105].

The Talairach coordinates of the studies included in the ALE meta-analysis were automatically converted into MNI coordinates using GingerALE. According to the modified procedures of Eickhoff and colleagues [98], the ALE values of each voxel in the brain were computed, and the null distribution of the ALE statistic was calculated for each voxel. The ALE method incorporates a variable uncertainty based on subject size. Since implementing the variable uncertainty of the random-effects method, GingerALE needs subject information for each foci group to calculate the Full-Width Half-Maximum(FWHM) of the Gaussian function used to blur the foci [98]. The thresholded ALE map was computed using p values from the previous phase, and we used a cluster forming threshold of *p* < 0.001 and a cluster-level threshold of *p* < 0.05 Familywise Error (FWE)corrected [106]. For each group (GD and AUD) activation (activation and deactivation), we performed separate ALE meta-analyses (Table 1 and Table 2).

## 5. Results

### 5.1. Neural Alterations in Gambling Disorder

The ALE meta-analysis on GD (Table 3) revealed clusters of activation in both hemispheres (Figure 2), mainly located in the head and body of the cuneus (CN; Figure 2), in the lentiform nucleus (LN), and in the hypothalamus (see Figure 2). We also found clusters of activation in the right middle frontal gyrus (MFG) and in the right putamen. No significant suprathreshold cluster of deactivation was detected in GD.

### 5.2. Neural Alterations in Alcohol Use Disorder

The ALE meta-analysis on AUD (Table 4 and Table 5) revealed clusters of activation in the lentiform nucleus (LN) including the superior portion of the left putamen (Figure 3). We also found clusters of activation in the right middle frontal gyrus (MFG), and in the left posterior cingulate cortex (PCC) extending to the precuneus. The ALE meta-analysis revealed clusters of deactivation in both hemispheres located in the MFG, and also the boundary between the middle and anterior portion of the left cingulate gyrus cortex (CGC) and the inferior portion of the left putamen (Figure 4).

## 6. Discussion

Executive functions encompass several cognitive processes, such as decision-making, response inhibition, conflict monitoring, cognitive flexibility, and their possible relationship with reward-related decision-making processes [55]. Executive functions have been repeatedly found to be altered in GD and AUD (see the “Introduction” section above). There are multiple reliable and valid computerized neurocognitive tasks used during fMRI scanning to assess neurofunctional alterations in GD and AUD; by including these tasks in our meta-analysis (see Table 1 and Table 2) and summarizing previous fMRI findings on AUD and GD, we demonstrated neurofunctional alterations in the brain network of executive control in individuals with AUD and GD. The present meta-analysis and results, going beyond the limitations of a single-study approach (e.g., low power and generalizability), provide, for the first time, broad and converging evidence that both disorders are associated with specific neural alterations in neural networks related to executive function. For ease of exposition, the discussion will be divided into subheadings.

### 6.1. Neurofunctional Alterations in the Executive Function Brain Network in GD

Our results revealed clusters of abnormal activation in GD bilaterally in the head and the body of the CN, in the right LN (including the putamen), and the right MFG. These results are in accordance with previous studies [18,22,33,107,108]. Individuals with GD, relative to HC, also showed increased activity in the left dorsal ACC. The brain reward system is crucial to survival by stimulating food consumption, social bonding, or reproduction through the release of dopamine in the NAc and frontal lobes. Our findings support the idea that some regions, such as the vs. and its afferent and efferent projections, are pathologically involved in “unnatural rewards”, leading to compulsive activities such as gambling, as occurs also for other types of addictions such as alcohol and nicotine [109].

The striatum has been widely reported to be involved in the subjectively discounted value of delayed rewards and in the anticipation of monetary rewards [31,48,54,83]. Indeed, individuals with GD showed greater activity in the bilateral dorsal striatum, linked to stronger action–outcome associations [27], which could be explained by an overestimation of the gambling outcomes. The hyperactivity of these regions appears to be associated with higher reward-seeking behavior, which could be a compensatory mechanism correlated to reward gaps in GD [27]. Instead, the vs. seems to be more involved in the processing of rewards per se [54]. The present findings showed higher activation in the LN during tasks engaging executive functions in individuals with GD, consistent with the idea that GD is a behavioral dependence associated with dysfunction in the reward circuit. The fronto-striatal cortical circuit (encompassing the right lenticular nucleus and the MFG) plays a pivotal role in executive functioning [110] and inhibitory control [111], including reward, control, and motor planning [112]. When the clinical syndromes are more severe, with higher SOGS scores, the patients with GD fail to regulate their gambling behavior due to the hyperactivity of the striatum. Thus, the functional alteration we detected here may contribute to fronto-striatal dysfunction in GD, which in turn results in abnormal self-regulation and the intense desire of gamblers to maximize the reward. 

In this vein, GD may arise from the imbalance between the dopaminergic system, including limbic motivational areas and neural pathways connecting the frontal lobe regions with the basal ganglia (fronto-striatal circuits), which results in impaired cognitive control, leading to progressive loss of control over gambling behaviors [55].

### 6.2. Neurofunctional Alterations in the Executive Function Brain Network in AUD

The present results revealed three clusters of activation in individuals with AUD, mainly located in the PCC extending to the precuneus, in the right MFG, and in the left putamen. The highest activation we detected in the left putamen may have been related to the computed value and/or hedonic elements of making an enhanced risky choice as monetary earnings in individuals with AUD. These areas receive both glutamatergic and dopaminergic inputs and have been involved in developing and maintaining alcohol dependence, consistent with behavioral evidence that alcohol-dependent individuals show a greater sensitivity to earning experiences in support of the future risky search for a reward. The greater activation we found in the occipital areas might be explained by the fact that those with AUD needed to recruit more resources in the visual area to inhibit processing of alcohol-related pictures [90]. We also found clusters of deactivation in the bilateral MFG, the cingulate cortex, and the left putamen. All of these areas work together and are involved in cognitive control, error processing, impulse control, and decision-making processes. Our results suggested that individuals with AUD have difficulties in adapting their responses to the changing contingencies of reward-guided decision-making. Weaker activity in the bilateral DLPFC and the dorsal anterior cingulate, associated with loss of adaptive control of action selection, could be linked with cognitive inflexibility when reinforcement contingencies change [42]. In Wrase and colleagues [19], individuals with AUD showed weaker sensitivity in the putamen and in the caudate head during the anticipation of monetary gain. Moreover, activity in the nucleus caudate was implicated in linking the reward to behavior and could be associated with an increased learning benefit from a reward in the HC [113].

### 6.3. Evidence for the Neurobiological Hypothesis of AUD and GD

The most accredited neurobiological hypothesis for addictive behavior is dysfunction in the “cortico-striatal reward pathway”, which includes areas such as the vs. and mPFC [114]. fMRI studies do not clarify the nature and direction of this dysfunction, with a number of studies showing activations and an almost equal number showing deactivations [115,116]. The converging evidence for activation in GD that we found here is in line with two seminal theories: the “incentive sensitization theory” and the “impulsivity theory”. The incentive sensitization theory posits an attentional bias towards cues related to addiction, thereby exacerbating compulsive pathological features [117]. This leads to an abnormal recruitment of the reward system, which results in a greater activation of the VS, especially when, as in GD, the reward is monetary. When incentive sensitization is combined with altered executive control, uncontrolled impulsiveness and sensation seeking, it can lead the individual to dependent behavior such as that observed in GD.

Our results also provide converging evidence for the deactivation in AUD of different brain circuits, including areas of the mPFC, the superolateral prefrontal cortex (sPFC), the ACC, and the striatum. These results are consistent with the theory of reward deficiency syndrome (“RDS”) [118], which highlights a general and chronic deactivation of reward circuits that reduces the pleasure of the gratification experience. These deactivated brain areas (striatum and mPFC), operating in a dysfunctional way, generate deficits in the monitoring of anticipation, expectancy, salience of rewards, and decision-making processes [119].

### 6.4. Conclusions and Future Directions

Here, we detected interesting findings about the direction of functional alterations in individuals with AUD and GD that have not been previously described. We found that GD was mainly associated with the higher activation of the fronto-striatal circuit (including the basal ganglia, MFG, and ACC). In contrast, AUD was associated with the higher activation of the occipital cortex and of the superior portion of the putamen, as well as with the deactivation of the fronto-striatal network (including the MFG, MCC, and inferior portion of the putamen). Thus, GD seems to be associated with an activation of the reward network, whereas AUD seems to be associated with both the deactivation and activation of the different nodes of this circuit. The direct comparison between the two conditions (namely, AUD and GD) is currently unfeasible due to the paucity of the studies meeting the inclusion criteria for an ALE meta-analysis. In addition, the state of the literature does not allow a clear distinction between the different executive processes involved, just as it is not possible to determine which is the most appropriate task to measure them. For such a reason, it has not been possible to distinguish between different sub-processes of the executive functions, because often the available studies do not describe what aspect of the executive functions has been investigated. This is due to the paucity of studies, and it represents both a limitation of the present study and the current modelling of executive function itself. Future studies should clarify and shed light on which aspects of the executive functions are really lacking in GD and AUD.

## Figures and Tables

**Figure 1 brainsci-10-00353-f001:**
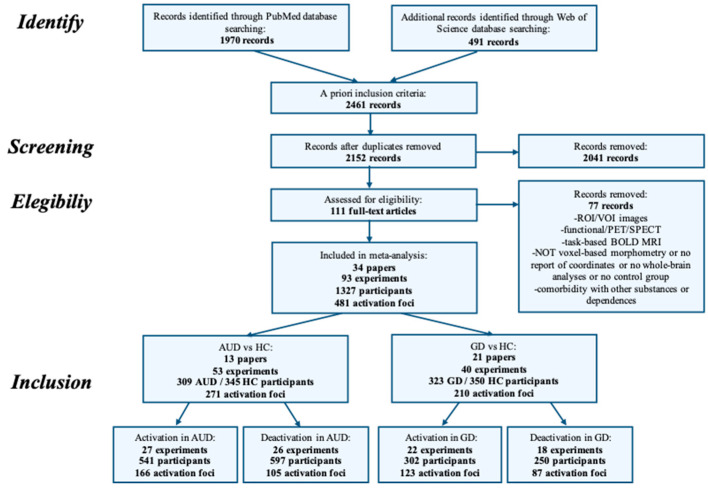
Flowchart illustrating the selection procedures for and details about the materials included in the meta-analysis. ROI = Region of Interest; VOI = Volume of Interest; PET = Positron Emission Tomography; SPECT = Single-Photon Emission Computed Tomography; BOLD = Blood Oxygenation Level Dependent; MRI = Magnetic Resonance Imaging; GD = Gambling Disorder; AUD = Alcohol Use Disorder; HC = Healthy Control.

**Figure 2 brainsci-10-00353-f002:**
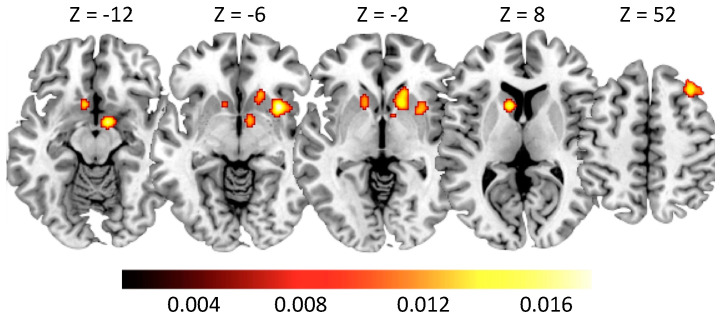
Clusters of activation in Gambling Disorder subjects. Montreal Neurological Institute (MNI) coordinates are provided. The color bar indicates activation likelihood estimation (ALE) values.

**Figure 3 brainsci-10-00353-f003:**
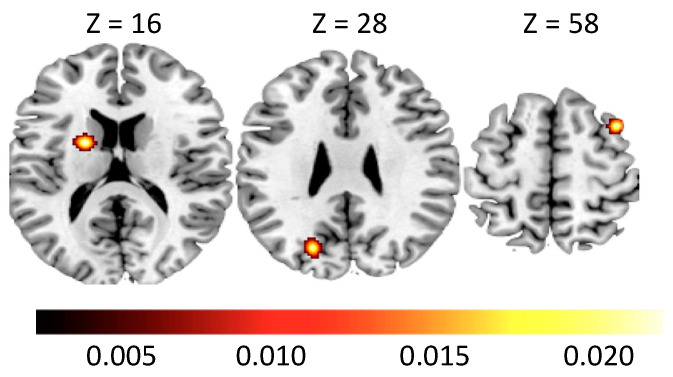
Clusters of activation in Alcohol Use Disorder subjects. MNI coordinates are provided. The color bar indicates ALE values.

**Figure 4 brainsci-10-00353-f004:**
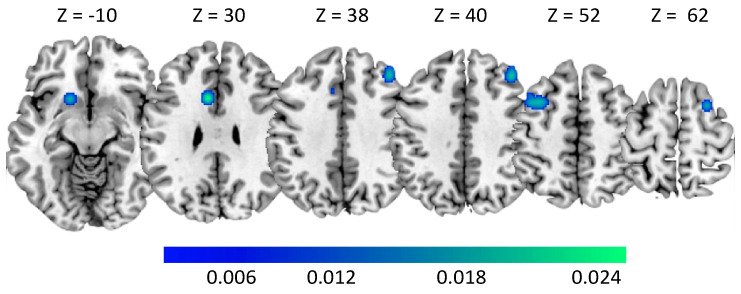
Clusters of deactivation in Alcohol Use Disorder subjects. MNI coordinates are provided. The color bar indicates ALE values.

**Table 1 brainsci-10-00353-t001:** Complete list of papers included in meta-analysis for Gambling Disorder. For each paper, details about the number of participants, the experimental paradigms, and the included contrasts in the studies are provided. GD = Gambling Disorder; HC = healthy control.

Gambling Disorder									
Paper	Subjects GD	Age M (SD)	Subjects HC	Age M (SD)	Experiments	Foci	Task	Enhanced Activation in GD (Relative to HC)	Reduced Activation in GD (Relative to HC)
Gelskov et al. 2016 [52]	14 (All male)	29.43 (6.05)	15 (All male)	29.87 (6.06)	3	27	Gambling	2	1
Brevers et al. 2015 [51]	10 (8 Male)	34.00 (8.53)	10 (8 Male)	36.20 (12.95)	2	2	Card-Deck Paradigm Task	-	2
Dixon et al. 2014 [76]	12 (All male)	-	10 (All Male)	-	2	16	Slot-Machine Task	2	-
Miedl et al. 2012 [54]	16 (15 male)	35.00 (2.00)	16 (15 male)	38.00 (2.00)	1	6	Delay discounting Task	1	-
Power et al. 2012 [48]	13 (All male)	42.40 (10.80)	13 (All male)	41.00 (11.00)	2	8	Iowa Gambling Task	1	1
Wiehler et al. 2017 [77]	24 (All Male)	29.68 (10.88)	24 (All male)	28.47 (7.13)	1	1	Delayed Monetary Reward Task	1	-
Sescousse et al. 2013 [78]	18 (All male)	34.10 (11.60)	20 (All male)	31.00 (7.30)	1	14	Monetary Incentive delay Task	1	-
Miedl et al. 2015 [79]	15 (All male)	36.70 (5.80)	15 (All male)	36.80 (5.60)	1	5	Monetary-choice Task	1	-
Miedl et al. 2010 [50]	12 (All male)	39.50 (9.30)	12 (All male)	33.40 (8.00)	3	11	Quasi-realistic Blackjack Task	2	1
Goudriaan et al. 2010 [20]	17 (All male)	35.30 (9.40)	17 (All male)	34.70 (9.70)	1	6	Cue Reactivity Task	1	-
Van Holst et al. 2012 [80]	15 (All male)	38.00 (13.42)	16 (All male)	34.92 (11.98)	3	7	Card guessing Task	3	-
Balodis et al. 2012 [81]	14 (10 male)	35.08 (11.70)	14 (10 male)	37.10 (11.30)	5	17	Monetary Incentive delay Task	-	5
Tsurumi et al. 2014 [82]	23 (All male)	32.6 (6.90)	27 (All male)	33.40 (8.00)	1	2	Monetary Incentive delay Task	-	1
Reuter et al. 2005 [31]	12 (All male)	37.30 (7.40)	12 (All male)	32.30 (5.60)	2	7	Card guessing Task	2	-
Crockford et al. 2005 [83]	10 (All male)	39.30 (7.60)	10 (All male)	39.20 (8.30)	1	3	Cue Reactivity Task	1	-
Habib et al. 2010 [84]	11 (10 male)	-	11 (4 male)	-	5	32	Computerize Slot-machine Task	2	3
Potenza et al. 2003 [33]	13 (All male)	31.15 (7.97)	11 (All male)	29.00 (7.81)	2	20	Stroop Task	1	1
Potenza et al. 2003 [53]	10 (All male)	36.20 (11.95)	11 (All male)	30.09 (7.71)	1	18	Video scenario gambling	-	1
Limbrick-Oldfield et al. 2017 [85]	20 (All male)	31.00 (range27-51)	22 (All male)	28.00 (range 25–52)	1	4	Cue reactivity Task	1	-
Fujino et al. 2018 [86]	23 (All male)	32.70 (7.80)	35 (All male)	29.30 (9.30)	1	3	Monetary Reward Task	-	1
Fujimoto et al. 2017 [87]	21 (All male)	34.70 (8.82)	29 (All male)	30.90 (10.40)	1	1	Goal-Instructed Gambling Task	-	1
**Total**	**323**		**350**		**40**	**210**			

**Table 2 brainsci-10-00353-t002:** Complete list of papers included in the meta-analysis for Alcohol Use Disorder. For each paper, details about the number of participants, the experimental paradigms, and the included contrasts in the studies are provided. AUD = Alcohol Use Disorder; HC = healthy control.

Alcohol Use Disorder									
Paper	Subjects AUD	Age M (SD)	Subjects HC	Age M (SD)	Experiments	Foci	Task	Enhanced Activation in AUD (Relative to HC)	Reduced Activation in AUD (Relative to HC)
Von Holst et al. 2014 [64]	18 (All male)	42.50 (10.40)	19 (All male)	40.40 (10.70)	3	6	Card guessing task	3	-
Gilman et al. 2015 [88]	18 (12 male)	31.20 (7.10)	18 (12 male)	30.50 (5.10)	5	13	Risk-taking Task	2	3
Jung et al. 2014 [89]	26 (All male)	50.20 (9.50)	26 (All male)	49.50 (11.40)	2	3	Decision-making Task	1	1
Beylergil et al. 2017 [42]	34 (All male)	44.73 (8.27)	26 (All male)	41.92 (9.59)	4	16	Reward-guided decision Task	-	4
Czapla et al. 2017 [90]	19 (17 male)	51.21 (7.36)	21 (17 male)	41.95 (9.99)	4	46	Go/No Go Task	3	1
Schulte et al. 2017 [91]	26 (18 male)	49.90 (9.50)	26 (17 male)	49.10 (11.00)	11	40	Alcohol Priming Stroop Task	5	6
Hu et al. 2015 [92]	24 (18 male)	38.70 (8.00)	70 (43 male)	35.10 (10.00)	2	9	Stop signal Task	1	1
Wrase et al. 2007 [19]	16 (All male)	42.38 (7.52)	16 (All male)	39.94 (8.59)	4	9	Monetary Incentive Delay Task	2	2
Li et al. 2009 [93]	24 (18 male)	38.70 (8.30)	24 (18 male)	35.50 (5.90)	3	17	Stop signal Task	1	2
Beck et al. 2009 [62]	19 (All male)	41.84 (6.79)	19 (All male)	41.68 (8.97)	3	13	Monetary Incentive Delay Task	-	3
Gilman et al. 2010 [94]	15 (8 male)	35.29 (7.34)	15 (8 male)	33.30 (8.60)	6	52	Judgment decision task	6	-
Sjoerds et al. 2013 [57]	31 (18 male)	48.50 (8.50)	19 (12 male)	47.70 (10.60)	4	6	Instrumental learning Task	2	2
Dennis et al. 2020 [95]	39 (29 male)	40.51 (9.09)	46 (28 male)	35.00 (11.80)	2	41	Probabilistic Discounting Task	1	1
**Total**	**309**		**345**		**53**	**271**			

**Table 3 brainsci-10-00353-t003:** Results of ALE meta-analysis on activation in Gambling Disorder. For each cluster region label, the hemisphere, ALE value, p and z values, and MNI coordinates are provided. LH = left hemisphere; RH = right hemisphere.

Cluster	Region	Hemisphere	ALE	*p*	Z	x	y	z
1	Caudate Head	RH	0.013	0.000	4.282	16	14	−2
	Caudate Head	RH	0.012	0.000	3.920	18	22	−2
2	Caudate Body	LH	0.014	0.000	4.428	−10	12	8
	Caudate Head	LH	0.011	0.000	3.806	−8	14	−12
	Caudate Head	LH	0.010	0.000	3.551	−14	14	−2
3	Putamen	RH	0.016	0.000	4.825	32	10	−6
4	Middle Frontal Gyrus	RH	0.013	0.000	4.241	34	24	52
5	Hypothalamus	RH	0.015	0.000	4.509	10	−2	−10

**Table 4 brainsci-10-00353-t004:** Results of ALE meta-analysis on activation in Alcohol Use Disorder. For each cluster region label hemisphere, the cluster size (mm^3^), ALE value, and MNI coordinates are provided. LH = left hemisphere; RH = right hemisphere.

Cluster	Region	Hemisphere	ALE	*p*	Z	x	y	z
1	Putamen	LH	0.021	0.000	5.149	−20	0	16
2	Middle Frontal gyrus	RH	0.020	0.000	5.008	38	12	58
3	Precuneus	LH	0.020	0.000	4.994	−20	−74	28

**Table 5 brainsci-10-00353-t005:** Results of ALE meta-analysis on deactivation in Alcohol Use Disorder. For each cluster region label hemisphere, the cluster size (mm^3^), ALE value, and MNI coordinates are provided. LH = left hemisphere; RH = right hemisphere.

Cluster	Region	Hemisphere	ALE	*p*	Z	x	y	z
1	Middle Frontal gyrus	LH	0.018	0.000	4.747	−34	10	52
	Middle Frontal gyrus	LH	0.017	0.000	4.721	−38	10	52
2	Middle Frontal gyrus	RH	0.019	0.000	5.028	40	34	40
3	Cingulate gyrus	LH	0.025	0.000	5.847	−8	14	30
	Cingulate gyrus	LH	0.010	0.000	3.383	−8	22	38
4	Putamen	LH	0.017	0.000	4.693	−16	14	−10
5	Middle Frontal gyrus	RH	0.018	0.000	4.867	24	8	64

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
