# Peer review of "Brain Network Underlying Executive Functions in Gambling and Alcohol Use Disorders: An Activation Likelihood Estimation Meta-Analysis of fMRI Studies"

_brainsci, 2020, doi:10.3390/brainsci10060353_

Round 1

Reviewer 1 Report

The authors have satisfactorily addressed my concerns.

Reviewer 2 Report

I am still skeptical about the inclusion of such an extreme variety of experimental paradigms under the label of “executive functions”, and I think that what the authors refer to as “different models” (Introduction) rather represent distinct tasks with completely different aims and instructions. However, these can still be labeled as “executive” insofar they share a control over the response.  Discussing this limitation is thus a reasonable compromise between selectivity and comprehensiveness.

I suggest a linguistic revision of the manuscript to correct some residual errors/typos, e.g. “nucleus caudate”.

This manuscript is a resubmission of an earlier submission. The following is a list of the peer review reports and author responses from that submission.

Round 1

Reviewer 1 Report

This work takes a quantitative meta-analytical approach to investigate the brain regions showing consistent hyper/hypo-activation in patients with gambling (GD) or alcohol-use (AUD) disorder, compared with controls, in association with tasks tapping "executive functioning". To this purpose, the authors performed an activation likelihood estimation meta-analysis of 34 (or 28?) fMRI studies addressing the neural bases of processes such as: response inhibition involving response execution (stop-signal task); b) response inhibition involving response selection (GoNogo task); c) selective attention/interference control (stroop task); d) reward sensitivity (several different tasks); e) intertemporal choice involving delayed gratifications (delay discounting task). Since the number of studies included in the meta-analysis did not allow formal direct comparisons (nor a conjunction analysis) between GD and AUD patients, only the results of group-specific ALE analyses were reported, showing: a) stronger activity, in GD patients compared with controls, in the basal ganglia, right middle frontal gyrus, right putamen and left anterior cingulate cortex; b) stronger activity, in AUD patients compared with controls, in the right cuneus extending to the middle occipital gyrus and in the superior portion of the left putamen; c) weaker activity, in AUD patients compared with controls, in the middle frontal gyrus bilaterally, left middle cingulate cortex and inferior portion of the left putamen. According to the authors, these findings provide novel evidence for specific neural alterations in the neural network for executive functions in both disorders.
While the goals of this work fit with the importance of novel insights into the neural bases of distinct types of addictions, there are major issues which unfortunately do not allow to recommend its publication, and which basically revolve around the numerosity of the included studies.

1) The first issue regards the extreme heterogeneity of the tasks used, and, most importantly, of the processes recruited, in the studies included in this meta-analysis (see above). It is largely acknowledged that the umbrella term "executive functioning" covers a variety of processes, ranging from response inbihition to decision-making (e.g. see Diamond 2013 on Annual Review of Psychology). Nevertheless, the aim of meta-analyses is to identify the brain areas associated with (families of) internally consistent *processes", with a considerable fine-grainess which cannot fit a whole continuum from response inhibition to intertemporal choice. One of such processes is the control/inhibition of reward-related appetitive and aversive drives, but their generation/update represents a completely different process, which could hardly be defined as executive: if the latter kind of process is to be considered "executive" because it entails a control over the response, then any "active" task used in neuroimaging settings should be considered as such.

2) Indeed, the name of the tasks used in the studies included in the meta-analysis (reported in Tables 1 and 2) are not useful in themselves, because their are not informative with regard to the processes which they recruited in the participants' brain. However, as detailed above these tasks clearly, and mostly, involve processes which can be related to executive functions only indirectly, i.e. to the extent that they entail behavioral drives. Apart from this, just to make an example it would be hard to classify the Iowa gambling task as an "executive" taks, if only because Damasio's Somatic marker hypothesis is grounded in the double dissociation between defective performance in this task after ventromedial prefrontal (but not dorsolateral prefrontal) impairment, and defective performance in "executive" tasks such as the WCST in the opposite case.

3) Related to the former issues (actually, one of their direct consequences), the manuscript mixes up few references to "executive functions" (particularly in the title and introduction) with a prominent discussion of notions revolving around the processing of reward (both Introduction and Discussion).

4) The above issues are likely to result from the paucity of studies addressing specifically "executive functions" (e.g. inhibitory control, working-memory, cognitive flexibility), which is the authors' account for the lack of direct group comparisons (and conjunction analyses) aimed to assess differential vs. common effects of GD and AUD on task-related brain activity. I understand this point, but at the same time it cannot be neglected that it is unusual to report, separately, the results of two unrelated (statistically speaking) meta-analyses. My feeling about this weakness is that there is simply insufficient evidence to publish a meta-analysis on this specific topic at this stage. Instead, the authors might consider to perform a meta-analysis on "decision-making" (see the 5th search criteria at p.6: "the experimental task required participants to perform a decision-making task or similar tasks that engage decisional processes"), but also in this case they would be expected to distinguish among distinct sub-processes within this huge domain (e.g. Liu et al., 2011 on Neuroscience and Biobehavioral reviews).

5) Related to the former issue, in the lack of explicit direct comparisons and conjunction analyses acros GD and AUD groups, it is not justified to claim "we provide an extensive revision of the current neuropsychological and neuroimaging literature on AUD and GD, aiming at providing a comprehensive picture of common and distinct features of the two conditions."

Other issues/suggestions:

6) abstract (and subsequent sections):
- it should be always clarified (legends included) that the results concern differences between GD/AUD and controls (i.e. stronger or weaker activity in patients vs. controls);
- the descriptions of results often start with an initial introduction to bilateral activations immediately followed by lateralized findings

7) it is a common procedure to report when (i.e. year and month/period) the search for studies has been performed

8) did the authors ensure that the same/similar tasks had been used in the studies performed by GD and AUD patients?

9) Figure1: the number of records removed after the "elegibility" stage might be missing;

10) Tables 1 and 2: the tables seem to be mixed up (maybe an error with formatting or saving in PDF?); moreover, what does the "Experiments" column report?

11) are statistics corrected for multiple comparisons? "The thresholded ALE map was computed using p values from the previous phase and a cluster-level inference at the 0.05 level of significance and a cluster forming threshold at p < 0.001 (uncorrected)"

12) What is the scale in the colorbars? It reports quite unsual values

13) p.10 "also found a cluster in the left cingulate gyrus cortex (CGC)"; the cingulate gyrus is a very extensive portion of the cortical midline, thus I would suggest to provide a more specific anatomical definition

14) p.13 "obtaining alcohol dependence" sounds odd to me.

15) p.13 "Moreover, activity in the putamen and caudate has been implicated in linking reward to behavior and could highlight a greater benefit from reward in HC but not in those with AUD." ; I would suggest to rephrase this sentence

16) Did the authors select 34 studies (abstract) or 28 (p. 6)?

Author Response

Response to Reviewer 1 Comments

1) The first issue regards the extreme heterogeneity of the tasks used, and, most importantly, of the processes recruited, in the studies included in this meta-analysis (see above). It is largely acknowledged that the umbrella term "executive functioning" covers a variety of processes, ranging from response inhibition to decision-making (e.g. see Diamond 2013 on Annual Review of Psychology). Nevertheless, the aim of meta-analyses is to identify the brain areas associated with (families of) internally consistent *processes", with a considerable fine-grainess which cannot fit a whole continuum from response inhibition to intertemporal choice. One of such processes is the control/inhibition of reward-related appetitive and aversive drives, but their generation/update represents a completely different process, which could hardly be defined as executive: if the latter kind of process is to be considered "executive" because it entails a control over the response, then any "active" task used in neuroimaging settings should be considered as such.

2) Indeed, the name of the tasks used in the studies included in the meta-analysis (reported in Tables 1 and 2) are not useful in themselves, because there are not informative with regard to the processes which they recruited in the participants' brain. However, as detailed above these tasks clearly, and mostly, involve processes which can be related to executive functions only indirectly, i.e. to the extent that they entail behavioral drives. Apart from this, just to make an example it would be hard to classify the Iowa gambling task as an "executive" tasks, if only because Damasio's Somatic marker hypothesis is grounded in the double dissociation between defective performance in this task after ventromedial prefrontal (but not dorsolateral prefrontal) impairment, and defective performance in "executive" tasks such as the WCST in the opposite case.

3) Related to the former issues (actually, one of their direct consequences), the manuscript mixes up few references to "executive functions" (particularly in the title and introduction) with a prominent discussion of notions revolving around the processing of reward (both Introduction and Discussion).

4) The above issues are likely to result from the paucity of studies addressing specifically "executive functions" (e.g. inhibitory control, working-memory, cognitive flexibility), which is the authors' account for the lack of direct group comparisons (and conjunction analyses) aimed to assess differential vs. common effects of GD and AUD on task-related brain activity. I understand this point, but at the same time it cannot be neglected that it is unusual to report, separately, the results of two unrelated (statistically speaking) meta-analyses. My feeling about this weakness is that there is simply insufficient evidence to publish a meta-analysis on this specific topic at this stage. Instead, the authors might consider to perform a meta-analysis on "decision-making" (see the 5th search criteria at p.6: "the experimental task required participants to perform a decision-making task or similar tasks that engage decisional processes"), but also in this case they would be expected to distinguish among distinct sub-processes within this huge domain (e.g. Liu et al., 2011 on Neuroscience and Biobehavioural reviews).

Response 1-4: First of all, we would like to thank the Reviewer for his/her comments and suggestions. We have considered the points 1 to 4 part of the same issue; indeed, executive functions (EF) are a “large umbrella” including several, different processes. We perfectly agree with the Reviewer, however, that, nowadays, fMRI studies on AUD and GD and executive functions do not allow a clear distinction between distinct EF processes. Accordingly, we modified the discussion adding a piece concerning this issue, also considering that the study we collected for the present meta-analyses do not allow to apply a clear distinction among different EF. This is a consequence of the paucity of fMRI studies on GD and AUD, and this could be both a limitation of the present study and the following current modelling of executive functions. Undoubtedly, future studies should clarify and shed some light on this aspect to make possible an operative distinction of this different aspects of executive functions in a meta-analysis study. Indeed, Muller et al. (2018) suggests that when planning a meta-analysis there always is the challenge to find a balance between homogeneity and power. In general, a meta-analysis aims to pool across different approaches and tasks in order to investigate effects consistent across strategies. The study by Muller et al (2018) provides an example of it: “a researcher interested in cognitive action control may want to know which regions are consistently found activated or deactivated across experiments that required participants to inhibit a prepotent response in favor of a non-routine one. For this example, the question arises if one should include all experiments that test cognitive action control, no matter what paradigm was used (e.g., Stop-signal, Go/No- Go, Stroop, Flanker tasks…), or limit the analysis to a specific paradigm (e.g., Stop-signal task). Considering the consequences for interpretation, the latter case would be specific to the cancelling of an already initiated action, while a meta-analysis across all paradigms would focus on the higher order supervisory control processes necessary in all paradigm types. Importantly, if one decides to include different paradigms, it may be helpful to ensure that the distribution of experiments is relatively balanced across tasks”. Basically, when you want to test “cognitive action control” independently of the paradigm used, you include different types of tasks because you are interested in that specific domain. These aspects have now been clarified in both Introduction and Discussion.

In the introduction section: “Present literature review suggests that the definition of “executive functions” includes a large umbrella of multiple processes and several, different definition of executive function exist, which refer to different cognitive and neuropsychological models. Therefore, we found that in studying executive functions in GD and in AUD authors referred to different models and, accordingly, used different tasks to analyze executive functions. The problem of the absence of a homogenous definition of executive function and that of the large variety used to assess them in clinical population has already been underlined in other meta-analyses (Kenrs et al.,2008)”

In the discussion section: “In addition, the state of the literature does not allow a clear distinction between the different executive processes involved, just as it is not possible to determine which is the most appropriate task to measure them. For such a reason, it has not been possible to distinguish between different sub-processes of the executive functions, because often the available studies do not describe what aspect of the executive functions has been investigated. This is due to the paucity of studies, and it represents both a limitation of the present study and the current modelling of executive function themselves. Future studies should clarify and shed some light on which aspects of the executive functions are really lacking in GD and AUD.”

5) Related to the former issue, in the lack of explicit direct comparisons and conjunction analyses across GD and AUD groups, it is not justified to claim "we provide an extensive revision of the current neuropsychological and neuroimaging literature on AUD and GD, aiming at providing a comprehensive picture of common and distinct features of the two conditions."

Response 5: We have rephrased this sentence: “we provide a revision of the current neuropsychological and neuroimaging literature on AUD and GD, evidencing possible differences or common features of the two conditions.”

Other issues/suggestions:

6) abstract (and subsequent sections): 
- it should be always clarified (legends included) that the results concern differences between GD/AUD and controls (i.e. stronger or weaker activity in patients vs. controls); 
- the descriptions of results often start with an initial introduction to bilateral activations immediately followed by lateralized findings.

Response 6: We thank the Reviewer for these useful suggestions, following them we have reformulated the abstract according to a description from general to the particular, specifying differences between groups.

7) it is a common procedure to report when (i.e. year and month/period) the search for studies has been performed

Response 7: We have reported the period in which we have collected the data: “the data included in this meta-analysis have been collected from early 2019 to February 2020”.

8) did the authors ensure that the same/similar tasks had been used in the studies performed by GD and AUD patients?

Response 8: The typology of tasks for each addiction are not perfectly balanced (1 to 1 ratio), however, we have discussed and clarified this in the “discussion section-limits”.

9) Figure1: the number of records removed after the "eligibility" stage might be missing.

Response 9: We thank Reviewer to have noticed this, unfortunately there was a formatting error in the flowchart (figure 1) that has produced several misunderstandings. We have now solved it, including in the Revised Manuscript the revised figure and tables.

10) Tables 1 and 2: the tables seem to be mixed up (maybe an error with formatting or saving in PDF?); moreover, what does the "Experiments" column report?

Response 10: Following the Reviewer’s suggestion, we have specified that the column “Experiment” in table 1 and 2 was referred to “number of experiments” for each study. The structure of both tables is intentionally the same except for the typology of addiction. We have added a first row in both tables that specifies which is the experimental group and therefore to which group the table refers.

11) are statistics corrected for multiple comparisons? "The thresholded ALE map was computed using p values from the previous phase and a cluster-level inference at the 0.05 level of significance and a cluster forming threshold at p < 0.001 (uncorrected)"

Response 11: We have now specified in the paper: “We used a cluster forming threshold of p < 0.001 and a cluster-level threshold of p < 0.05 FWE-corrected”.

12) What is the scale in the colorbars? It reports quite unsual values

Response 12: The color bar indicates the ALE value. (See the study of Eickhoff et al., 2009). We have now reported this detail in the figure caption.

13) p.10 "also found a cluster in the left cingulate gyrus cortex (CGC)"; the cingulate gyrus is a very extensive portion of the cortical midline; thus I would suggest providing a more specific anatomical definition.

Response 13: We have now provided a more specific definition of cingulate gyrus cortex (CGC), in: “boundary between middle and anterior portion of left cingulate gyrus cortex (CGC)”.

14) p.13 "obtaining alcohol dependence" sounds odd to me.

Response 14: We have modified “obtaining” with “developing”.

15) p.13 "Moreover, activity in the putamen and caudate has been implicated in linking reward to behavior and could highlight a greater benefit from reward in HC but not in those with AUD."; I would suggest rephrasing this sentence

Response 15: We have rephrased this sentence: “Moreover, activity in the nucleus caudate was implicated in linking reward to behaviour and could be associated with increased learning benefit from reward in HC (Knutson et al., 2005)”.

16) Did the authors select 34 studies (abstract) or 28 (p. 6)?

Response 16: We thank Reviewer for his question that allows us to clarify this point and to correct it in the abstract. The studies selected were 34. We have modified the uncorrected reported number that appeared in the abstract.

Reviewer 2 Report

Review for manuscript brainsci-789660-v1 “Brain network underlying executive functions in 2 gambling and alcohol use disorders: an activation 3 likelihood estimation meta-analysis of fMRI studies” by Quaglieri et al.

            This manuscript reports a review and meta-analysis of neuroimaging and neuropsychological studies of gambling disorder (GD) and alcohol use disorder (AUD). The purpose of this study was to examine similarities and differences between these two disorders in terms of neural correlates of executive functions. The authors reviewed multiple studies of GD and AUD before performing an activation likelihood estimation meta-analysis of a subset of the reviewed studies. This subset was chosen according to a variety of a priori inclusion criteria chosen in accordance with PRISMA Statement guidelines. The study found that GD was associated with bilateral activation of the basal ganglia, right middle frontal gyrus, right putamen, left ACC, and subcortical regions such as the cuneus, LN, and hypothalamus. AUD was found to be associated with activation of the LN, right cuneus, middle frontal gyrus, and superior left putamen, and deactivation of bilateral middle frontal gyrus, left cingulate cortex and inferior left putamen. The authors conclude that both disorders are associated with alterations in the neural network for executive functions.

I found this study to be a very interesting comparison of two disorders that putatively involved impairments in executive functioning. The study findings appear to confirm this hypothesis and provide useful details about the executive control-related neural circuits that are involved in these disorders. That said, I do have some concerns about the meta-analysis and its conclusions, which are listed below:

1) lines 225 – 255: What were the terms used for the electronic literature search? Examples of these terms should be provided to the reader.

2) lines 225 – 255: how many of the authors abstracted the data from the included studies? Was this data abstract independent across authors? Were there any disagreements between authors on including/excluding studies? What were those disagreements and how were they resolved? This information should also be provided to the reader.

3) lines 225 – 255: The authors need to double-check the numbers in Figure 1, as many of them are not consistent with each other. For example, in the eligibility section of the figure, the authors report 481 activation foci in the meta-analysis, but then report 271 (AUD vs HC) + 209 (GD vs HC) = 480 activation foci in the Inclusion section of the figure. Moreover, the Eligibility section of the figure is confusing; it is unclear what the text entries in the right column of this figure are referring to. Also, the heading for this section is misspelled, and the formatting for this section (Right text column) needs to be improved (e.g. the word “dependencies” is isolated from the rest if its sentence).

4) lines 225 – 255: Table 2 is very useful, but subject demographics (e.g. age, gender, etc.) should also be reported. The authors should also double check the consistency of the numbers for this table too.

5) lines 256 – 314: Is it possible to include some metric of uncertainty for the ALE statistics, such as 95% confidence intervals?

6) lines 256 – 314: How did the authors assess publication bias? The authors should report this information as well.

7) lines 284 – 314: There is an inconsistency in the areas of activation for GD and AUD as reported in the Results sections and the manuscript Abstract (lines 22 – 25). The authors should double-check and correct as needed.

8) Lines 334 – 336: In the Discussion section, the authors state that “[o]ur findings support the idea that some regions, such as the VS and its afferent and efferent 334 projections, are pathologically involved in “unnatural rewards”, leading to compulsive activities such as gambling, as occurs also for other types of addictions, such as alcohol and nicotine”. I assume the acronym “VS” means “Ventral Striatum”? I assume this nomenclature is used because the dorsal striatum consists of the caudate nucleus and the putamen, two regions that were activated/deactivated in the AUD group. It would be good if the authors made this identification clearer for readers who do not have a knowledgeable background in neuroanatomy.

9) In general, it would also be helpful to the reader if the authors ensured that all acronyms used in this paper were clearly defined (for example, the acronym “HC”, which I assume means “healthy controls”).

Author Response

Response to Reviewer 2 Comments

I found this study to be a very interesting comparison of two disorders that putatively involved impairments in executive functioning. The study findings appear to confirm this hypothesis and provide useful details about the executive control-related neural circuits that are involved in these disorders. That said, I do have some concerns about the meta-analysis and its conclusions, which are listed below:

1) lines 225 – 255: What were the terms used for the electronic literature search? Examples of these terms should be provided to the reader.

Response 1: The terms used for the electronic literature search have been showed in paragraph “Inclusion criteria for papers”: “fMRI” AND “gambling” OR “alcohol” AND “decision-making” AND “gambling disorder” OR “alcohol abuse disorder” OR “MRI” OR “functional MRI”.

2) lines 225 - 255: how many of the authors abstracted the data from the included studies? Was this data abstract independent across authors? Were there any disagreements between authors on including/excluding studies? What were those disagreements and how were they resolved? This information should also be provided to the reader.

Response 2: Studies included and data extraction in this meta-analysis were screened in independent double-checking following the PRISMA guidelines (see the PRISMA checklist by Liberati et al., 2009). We have provided a description process in the text: “Independent double-checking was used to avoid the biases in the data extraction. In addition, the same independent investigators have provided the correctness of the space (MNI or Talairach) and the coordinates”.

3) lines 225 - 255: The authors need to double-check the numbers in Figure 1, as many of them are not consistent with each other. For example, in the eligibility section of the figure, the authors report 481 activation foci in the meta-analysis, but then report 271 (AUD vs HC) + 209 (GD vs HC) = 480 activation foci in the Inclusion section of the figure. Moreover, the Eligibility section of the figure is confusing; it is unclear what the text entries in the right column of this figure are referring to. Also, the heading for this section is misspelled, and the formatting for this section (Right text column) needs to be improved (e.g. the word “dependencies” is isolated from the rest if its sentence).

Response 3: We thank Reviewer to have notice this, unfortunately, the figure about flowchart had visualization problems (upload/formatting problem), that have now been solved. Regarding the number of activation foci, we have checked and corrected the number of missing activation foci (271 AUD activation foci) + (210 GD activation foci) = 481.

4) lines 225 – 255: Table 2 is very useful, but subject demographics (e.g. age, gender, etc.) should also be reported. The authors should also double check the consistency of the numbers for this table too.

Response 4: We have modified the table 1 and table 2 providing demographics descriptions (i.e., age and gender), except for except for two studies which did not report such data.

5) lines 256 – 314: Is it possible to include some metric of uncertainty for the ALE statistics, such as 95% confidence intervals?

Response 5: Following the Reviewer’s suggestion, we have now included in “Activation Likelihood Estimation” more detailed information about metric of uncertainty in ALE statistics. We have added in the text: “ALE method incorporates a variable uncertainty based on subject size. Since implementing the variable uncertainty of the random‐effects method GingerALE needs subject information for each foci group to calculate the Full‐Width Half‐Maximum (FWHM) of the Gaussian function used to blur the foci (Eickhoff et al. 2009).

 6) lines 256 – 314: How did the authors assess publication bias? The authors should report this information as well.

Response 6: Differently from the effect-size meta-analysis the neuroimaging meta-analysis allow to test for spatial convergence of effects across experiments (Muller et al., 2018). Moreover, the strict inclusion criteria of the study defined at the beginning allows to check and control for some of these biases.

7) lines 284 – 314: There is an inconsistency in the areas of activation for GD and AUD as reported in the Results sections and the manuscript Abstract (lines 22 – 25). The authors should double-check and correct as needed.

Response 7: We thank the Reviewer for his/her suggestion, the inconsistency in the abstract has been now modified. We have aligned the abstract with the results section.

8) Lines 334 – 336: In the Discussion section, the authors state that “[o]ur findings support the idea that some regions, such as the VS and its afferent and efferent 334 projections, are pathologically involved in “unnatural rewards”, leading to compulsive activities such as gambling, as occurs also for other types of addictions, such as alcohol and nicotine”. I assume the acronym “VS” means “Ventral Striatum”? I assume this nomenclature is used because the dorsal striatum consists of the caudate nucleus and the putamen, two regions that were activated/deactivated in the AUD group. It would be good if the authors made this identification clearer for readers who do not have a knowledgeable background in neuroanatomy.

Response 8: We confirmed that “VS” means ventral striatum, we have provided a specification of this region: “VS, including nucleus accumbens and olfactory tubercle”.

9) In general, it would also be helpful to the reader if the authors ensured that all acronyms used in this paper were clearly defined (for example, the acronym “HC”, which I assume means (“healthy controls”).

Response 9: We thank the Reviewer for his/her advice, we have now reviewed the entire manuscript and provided a first full description of the acronyms and then we have provided an identification with round bracket.
